# Combined Serum ALBUMIN with Neutrophil-to-Lymphocyte Ratio Predicts the Prognosis of Biliary Tract Cancer after Curative Resection

**DOI:** 10.3390/cancers15225474

**Published:** 2023-11-19

**Authors:** Tai-Jan Chiu, Yueh-Wei Liu, Chee-Chien Yong, Shih-Min Yin, Cheng-His Yeh, Yen-Yang Chen

**Affiliations:** 1Division of Hematology-Oncology, Department of Internal Medicine, Kaohsiung Chang Gung Memorial Hospital, Taiwan and Chang Gung University College of Medicine, Kaohsiung 833, Taiwan; chiutaijan@gmail.com; 2Kaohsiung Chang Gung Cholangiocarcinoma and Pancreatic Cancer Group, Cancer Center, Kaohsiung Chang Gung Memorial Hospital, Kaohsiung 833, Taiwan; anthony0612@cgmh.org.tw (Y.-W.L.); ccyong3980@gmail.com (C.-C.Y.); b9302069@cgmh.org.tw (S.-M.Y.); ycc9002108@cgmh.org.tw (C.-H.Y.); 3Department of Surgery, Kaohsiung Chang Gung Memorial Hospital, Kaohsiung 833, Taiwan

**Keywords:** albumin, neutrophil-to-lymphocyte ratio, biliary tract cancer, surgery, prognosis

## Abstract

**Simple Summary:**

More than 60% of biliary tract cancer patients receive curative surgical resection. Nethereless, there is an absence of useful tools to predict prognosis before operation. Serum albumin and neutrophil-to-lymphocyte ratios are easily determined. Serum albumin is a marker of nutrition status and is correlated with prognosis in many cancer treatments. A neutrophil-to-lymphocyte ratio is a novel parameter widely used in most medical fields because it is a reliable marker of the immune response to different stimuli. Our study demonstrates that combined albumin and neutrophil scores, including nutritional and inflammatory indices, could predict operative survival in resectable biliary tract cancer patients.

**Abstract:**

Background: The mainstay treatment of biliary tract cancer is complete tumor resection. Prior to surgery, risk stratification may help to predict and plan treatment approaches. In this study, we investigated the possibility of combining serum albumin concentrations and neutrophil-to-lymphocyte ratios (NLR) to create a score as ANS to predict the prognoses of biliary tract cancer before surgery. Methods: This study retrospectively collected serum albumin concentration, neutrophil, and lymphocyte data measured in biliary tract cancer patients slated to receive complete tumor resections within two weeks before surgery. From January 2013 to December 2019, 268 biliary tract cancer patients who had received tumor resections at our hospital were categorized into 3 ANS groups: ANS = 0 (high albumin and low NLR), ANS = 1 (low albumin or high NLR), and ANS = 2 (low albumin and high NLR). Results: Five-year survival rates were 70.1%, 47.6%, and 30.8% in the ANS = 0, 1, and 2 groups, respectively. The median overall survival time for the ANS = 0 group could not be determined by the end of the study, while those for ANS = 1 and ANS = 2 groups were 54.90 months and 16.62 months, respectively. The results of our multivariate analysis revealed that ANS could be used as an independent predictor of overall and recurrent-free survival. A high ANS was also correlated with other poor prognostic factors. Conclusions: The ANS devised for this study can be used to predict postoperative survival in patients with BTC and to guide treatment strategies.

## 1. Introduction

Biliary tract cancers (BTC), including intrahepatic, perihilar, distal cholangiocarcinoma, and gallbladder cancer, are a spectrum of relatively low-incidence and highly aggressive malignancies [1]. BTC comprises 4% of all gastrointestinal tract cancers and is the second most frequent hepatobiliary cancer [2]. Intrahepatic and extrahepatic cholangiocarcinomas are reported to be more common in males, though the incidence of gallbladder cancer is known to be two- to six-fold higher in women. Despite its rarity, the incidence of BTC has increased yearly [3]. The prognosis of BTC is dismal in most cases because it is often diagnosed when it has already reached an advanced unresectable status. BTC outcomes are also adversely affected by tumor aggressiveness and ineffective therapeutic options [4]. Despite advances in surgical techniques and adjuvant chemotherapy, the recurrence rate remains high for patients with localized BTC, even after complete surgical resection [5,6]. 

TNM tumor staging is essential for predicting clinical outcomes and making decisions regarding adjuvant treatment. However, prognosis varies greatly, even in patients at the same disease stage. Thus, it would be useful to find an accurate preoperative prognostic factor to better assess the risk–benefit and provide patients with more personalized treatment. Some studies have found associations between BTC prognosis and specific molecular biomarkers involved in angiogenesis, cell mutation, proliferation, and differentiation. Excision repair cross complementation group 1 (ERCC1) is encoded for the nucleotide excision repair (NER) complex, and high ERCC1 expression has been shown to predict a poor prognosis in different types of cancer, including biliary tract cancer [7,8,9]. X-ray repair cross-complementing group 1 (XRCC1), a gene encoding a scaffolding protein for the base excision repair (BER) system, is reported to be a predictive marker in chemotherapy treatment and prognosis of biliary tract cancer [10]. However, these biomarkers require expensive laboratory techniques and comprehensive tests. It would be more practical to identify some simple serum biomarkers to help clinicians assess prognosis, adopt preventive measures, and decide on therapeutic strategies for patients with resectable BTC.

The prognostic nutritional indexes are markers of nutritional status and systemic inflammation based on serum albumin concentration and blood white cell counts, which can be easily obtained from routine preoperative blood tests. Albumin is a marker of nutrition status, and hypoalbuminemia was frequently observed in advanced cancer patients and predicted poor survival, including BTC [11]. Systemic inflammatory responses have been correlated with carcinogenesis, tumor proliferation, and migration [12]. Those studied in relation to BTC prognosis include the Glasgow prognostic score (GPS), the neutrophil–lymphocyte ratio (NLR), and the platelet–lymphocyte ratio (PLR) [13,14,15]. Some studies have explored the prognostic accuracy of combining different biliary tract cancer biomarkers, including the blood neutrophil-to-lymphocyte ratio and γ-glutamyl transpeptidase-to-platelet ratio index in intrahepatic cholangiocarcinoma [16]. The Prognostic Nutritional Index (PNI) and the Lymphocyte-to-Monocyte ratio (LMR) were implicated in the evaluation of outcomes of cancer patients [17]. However, limited studies have evaluated the prognostic factors of combined biomarkers of preoperative albumin and inflammation in biliary tract cancer patients post-operation. 

Therefore, we investigated the possibility of using a composite index score (ANS) derived from preoperative serum albumin levels and NLR to predict disease-free survival and overall survival in biliary tract cancer patients slated to receive tumor resections. 

## 2. Materials and Methods

### 2.1. Patient Selection

This study retrospectively enrolled biliary tract cancer patients who had received tumor resections at Kaohsiung Chang Gung Memorial Hospital between January 2013 and December 2019. Patients were excluded if they did not have complete laboratory data or if they were receiving operations for tumor recurrence and distant metastases. This study included a total of 287 biliary tract cancer patients, all of whom had pathologically proven adenocarcinoma. Biliary tract cancers were staged according to the seventh edition of the American Joint Committee on Cancer TNM staging system. After the operation, the cases of all patients were discussed in a multidisciplinary meeting of surgeons, medical oncologists, radiation oncologists, pathologists, and radiologists. According to current guidelines, adjuvant chemotherapy is suggested for patients with regional lymph nodes or unclear surgical margins if there are no medical contraindications. 

This study complied with the standards set forth in the Declaration of Helsinki and was approved by the Ethical Committee of Chang Gung Memorial Hospital. 

### 2.2. Data Collection

We collected demographic and clinical laboratory data, including patients’ age, gender, Eastern Cooperative Oncology Group performance status (ECOG PS), hepatitis B and C markers, serum concentrations of CRP, albumin, CA19-9 lymphocyte, and neutrophil count from the medical records two weeks before the operation. Post-surgery data included tumor stage according to the 8th AJCC staging manual and the pathological results, including surgical margin, vascular invasion, and adjuvant treatments. Adjuvant chemotherapy consisted of 5-FU- or gemcitabine-based chemotherapy. 

This study also tested the predictive value of two DNA-repair proteins (ERCC1 and XRCC1) in our biliary tract cancer patients. The resection specimens were retrieved from the Department of Pathology of Kaohsiung Chang-Gung Memorial Hospital and reviewed. A standard protocol was used for IHC staining of a monoclonal ERCC1 antibody (Thermo Scientific monoclonal Cat# ms-671-p0 antigen retrieval Tris-EDTA, pH9, 1/3000 1 h, clone 8F1) and a monoclonal XRCC1 antibody (Thermo Scientific monoclonal Cat# ms-434-po antigen retrieval Tris-EDTA, pH9, 1/2000 1 h, clone 32-2-5). Immunostaining was evaluated by experienced pathologists blinded to the clinical data. Tumor cells that showed distinct nuclear or cytoplasmic staining of more than 10% of the cells were considered positive [9,10].

The patients were regularly followed up at 3- to 6-month intervals, at which time they received laboratory tests, abdominal dynamically computed tomography scans, and sonography of the abdomen. Patients’ survival outcomes were also recorded. Progression-free survival (PFS) was defined as the time from tumor resection to disease progression or death due to any cause. Overall survival (OS) was the time from surgery until death or the last follow-up. NLR was calculated as the absolute neutrophil count divided by the absolute lymphocyte count. Albumin and neutrophil-to-lymphocyte ratio score (ANS) was calculated by assigning 0 or 1 point for albumin and NLR levels above or below the cut-off value, producing an ANS ranging from 0 to 2. The cut-off was set as the median value. 

### 2.3. Statistical Analysis

Data were presented as whole numbers with percentages for categorical variables. The chi-square and Wilcoxon tests were used to compare clinicopathological characteristics in biliary tract cancers. All patients received follow-up evaluations at our outpatient clinic until December 2020 or death. Unadjusted and multivariate analyses were performed to identify potential prognostic factors. Variables significantly associated with survival at the 0.05 level in the unadjusted analysis were considered for inclusion into a multivariate Cox proportional hazards model. All data were analyzed using SPSS statistical software package, version 19.0 (IBM Corporation, Armonk, NY, USA). Alpha was set at 0.05, and all tests were two-tailed.

## 3. Results

### 3.1. Patient Characteristics and Clinicopathological Data

Of the 267 biliary tract cancer patients, 167 (58.2%) were male and 120 (41.8%) were female. The median age was 61.5 years old (ranging from 31 to 86 years old), and the median follow-up period was 34.6 months. The primary sites of biliary tract cancer included intrahepatic (61.7%), gall bladder (12.9%), common bile duct (17.8%), and perihilar (7.7%). A total of 99 patients (34.5%) had chronic hepatitis B infections, and 48 (16.7%) had hepatitis C infections. The median CA19-9 value of all patients one month before surgery was 350 (2–50,000) U/mL. The pathological studies of surgical biopsies revealed that 83 patients (28.9%) had stage I lesions, 124 patients (43.2%) had stage II lesions, 39 patients (13.6%) had stage III lesions, and 41 patients (14.3%) had stage IV lesions. One hundred thirty-five patients (47%) did not receive lymph node dissections. A total of 37 patients (12.9%) received R1 resections, and 33 patients (11.5%) had close surgical margins. A total of 73 patients (25.4%) received adjuvant treatments: 53 adjuvant chemotherapy, 7 adjuvant radiotherapy, and 13 concurrent chemoradiotherapy (CCRT) (Table 1). 

### 3.2. Prognostic Factors of Resected Biliary Tract Cancer Patients

The dichotomization of patients by albumin and NLR levels was performed using the cuff-off value, which was 3.5 g/L for albumin and 2.28 for NLR. The optimal thresholds for NLR were defined using the receiver operating characteristic (ROC) curve analyses. To calculate the ANS, values below the cut-off value for albumin and over the cut-off value for NLR were given 1 point each, giving each patient a score from 0 to 2. The ANS score was 0 for 23 patients (8%), 1 for 162 patients (56.4%), and 2 for 102 patients (35.5%). A high ANS was associated with primary tumor locations, advanced tumor stage, vascular invasion, unclear surgical margin, HBV infection, high CA19-9 expression, and poor ECOG performance status (Table 2). The ANS score was also highly associated with the recurrence rate (Table 2), which was 34.4% (33/96) in patients with ANS = 0, 50.8% (64/126) in patients with AN = 1, and 67.7% (44/65) in those with ANS = 2 (Figure 1). 

### 3.3. Recurrence-Free Survival and Overall Survival 

At the end of this retrospective study, 141 (49.1%) biliary tract cancer patients experienced tumor recurrence after the operation, and 122 (42.5%) patients died. The median recurrence-free survival time was 35.1 months (Figure 2). In univariate analyses of recurrence-free survival of resected biliary tract cancer patients, poor ECOG performance status, advanced tumor stage, more lymph node metastases, stage III/IV AJCC pathological tumor, vascular invasion, R1 tumor resection, adjuvant treatments, hepatitis B infections, CA19-9 **≧** 350 U/mL, albumin <3.5 g/dL, and high ANS were linked to significantly shorter recurrence-free survival time (Table 3). At the multi-variates analyses, advanced tumor stage and lymph node metastases, vascular invasion, high CA19-9, and ANC = 2 showed significantly poor prognoses after tumor resection (HR: 0.497, 0.30–0.82; *p* = 0.007) (Table 3). The median RFS time for ANS group 0 was not calculated at the end of the study; the median RFS time for ANS group 1 was 35.12 months, and that of ANS group 2 was 9.40 months (Figure 3A). 

By the end of the study, median overall survival was 87.0 months (Figure 2). Our unadjusted analyses found shorter overall survival to be significantly associated with ECOG performance status scores of 0 and 1, common bile duct/hilar cholangiocarcinoma, pathological proved stage 3/4 tumors, lymph node metastases, AJCC tumor stage III/IV, R1 surgical margin, vascular invasion, chronic hepatitis B infection, CA19-9 **≧** 350, albumin <3.5, NLR **≧** 2.28, and ANS = 1 and ANS = 2 (Table 4). Our multivariate Cox-regressive survival analyses found a poor overall survival prognosis to be associated with AJCC tumor stage III/IV, vascular invasion, R1 surgical margin, adjuvant treatment, high CA19-9, and ANS = 2. The five-year survival rate was 70.1%. 47/6%, and 30.8% in ANS = 0, ANS = 1, and ANS = 2 groups, respectively. The median OS for the ANS = 0 group could not be calculated by the end of the study. It was 54.90 months and 16.62 months for the ANS = 1 and ANS = 2 groups, respectively (Figure 3B). 

We postulated that the ANS score based on the preoperative serum albumin concentration and the NLR might be a promising tool for predicting outcomes in BTC patients. To further evaluate the discriminatory ability of the prognostic scores, we compared the AUCs of albumin, NLR, and ANS (Table 5, Figure 4A,B). The ANS score had a higher AUC value than those of albumin and NLR, suggesting that the ANS might be a better marker of systemic inflammation and malnutrition than serum albumin and NLR alone.

## 4. Discussion

After surgical tumor resection, the pathology-determined tumor stage is often used to predict post-surgical outcomes in people with biliary tract cancer. However, this staging considers tumor characteristics only and does not consider other biological factors. To date, there have been no easily assessed pre-surgical clinical biomarkers found that are able to predict prognosis in biliary tract cancer patients after surgery. In this retrospective study, we investigated the possibility of using an index (ANS) combining a nutritional marker, albumin, and a systemic inflammation marker, NLR, to achieve this. We found a strong correlation between a high ANS score (2, indicating both low albumin concentrations and high pre-surgical NLR) and features of aggressive tumors, including advanced tumor stage, regional lymph node metastases, unclear surgical margin, hepatitis B infection, poor performance status, and high CA19-9. In addition, hypoalbuminemia and higher NLR also resulted in worse recurrent-free survival and overall survival. Our study demonstrated the efficacy of the combined systemic inflammation marker NLR and nutritional marker albumin (ANS score) in stratifying risks of BTC patients before surgery into three groups. The overall recurrence rate of this study was 49.1%: in ANS group 0, it was 34.4%; in ANS group 1, it was 50.8%; and in ANS group 2, it was 67.7%. Moreover, the overall survival rate in this study was 42.9%: in ANS group 0, it was 27.1%; in ANS group 1, it was 46.0%; and in ANS group 2, it was 60.0%. BTC patients in the high ANS group were associated with poor clinical and pathological factors that predicted poorer survival than those in the low groups. The result suggests that ANS is effective in helping clinicians determine BTC patient outcomes before making treatment policies in addition to TNM classification. 

In the literature review, the ANS score showed its efficacy in predicting outcomes in patients with oral squamous cell carcinoma, gastric adenocarcinoma, and pancreatic cancer [18,19,20]. Patients in higher ANS groups with oral squamous cell carcinoma also correlated with poor prognostic factors, such as higher pathological tumor TMN stage, perineural invasion, extranodal extension, and deeper tumor depth after the operation. In multivariate analyses, oral squamous cell carcinoma patients in ANS groups 1 and 2 also had poorer overall survival than ANS group 0. In gastric cancer, a high serum albumin-NLR score was an independent predictor of overall survival other than the TNM tumor stage. It was significantly associated with a higher TNM stage, more metastatic lymph nodes, and larger tumor size in resectable gastric cancer. These findings in oral squamous cell carcinoma and gastric were identical to our current research. In advanced or metastatic pancreatic ductal adenocarcinoma, ANS was an independent variable for overall survival and time-to-treatment failure in multivariate analyses. In this study, patients in the ANS 2 group had significantly higher incidences of chemotherapy-related toxicities than those in the other two groups. This study demonstrated that the ANS was an independent prognosticator in pancreatic cancer patients receiving chemotherapy.

Several studies showed that a higher NLR was correlated with a poor prognosis of resection of intrahepatic cholagniocarcinoma [21], gallbladder cancer [22], hilar cholangiocarcinoma [23], and distal cholangiocarcinoma [24]. In addition, a higher preoperative serum albumin level is associated with better long-term survival in intrahepatic cholangiocarcinoma and hilar cholangiocarcinoma patients [25,26]. The Glasgow prognostic score demonstrated that albumin combined with CRP had predictive value in preoperative BTC cancer patients [13]. Although these inflammation and nutrition markers were used individually to predict prognosis in preoperative BTC patients, only a few studies have explored the combination of markers to evaluate preoperative patients. 

The relationship between inflammation and cancer has been explored [27]. Inflammation predisposes cancer development and promotes all stages of tumorigenesis [28]. Numerous previous studies have reported serum systemic inflammatory markers that can be useful for predicting prognosis. Different systemic inflammatory indexes, such as the Glasgow prognostic score, NLR, and PLR in resectable BTC patients, have been demonstrated to predict outcomes effectively [7,17]. Studies from Muller et al. and Salati et al. showed that NLR and serum albumin in ALAN scores were correlated with prognoses in advanced unresectable BTC patients [29,30]. The NLR regards the balance between systemic immunity and inflammation and is associated with survival in different malignancies, such as the head/neck, liver, stomach, esophagus, and breast [31,32,33,34]. Neutrophils mediate proangiogenic factors and regulate inflammatory cytokines IL-1β in multistage carcinogenesis [35,36]. Lymphocytes are involved in the innate and adaptive immune response to eradicate tumor cells through cytotoxic cell death and cytokine secretion [37]. Higher neutrophil expression may inhibit the cytolytic function of lymphocytes and natural killer cells to cancer cells and cause poor prognosis [38]. In our study, high NLR was highly correlated with advanced tumor stage, lymph node metastases, unclear surgical margin, diabetes mellitus, viral hepatitis B infection, poor performance status, and early tumor recurrence. It also predicted shorter recurrence-free survival and overall survival. These findings were similar to Fei et al.’s study on resectable gallbladder cancer [39].

Serum albumin concentration often decreases during acute inflammation, correlated with poor prognosis in sepsis, heart failure, renal disease, and cancers [40,41,42]. Cancer-related cachexia and increased vascular permeability inducing albumin shift also contribute to hypoalbuminemia [43,44]. In some epidemiological studies, low serum albumin was correlated with cancer-related mortalities [45]. Hypoalbuminemia has prognostic values in several malignancies, including colorectal, endometrial, ovarian, and vulvar cancers [46,47,48,49]. In the literature review, only a few studies explored the role of preoperative hypoalbuminemia in resected biliary tract cancer [13,50]. Our study showed that preoperative serum albumin was strongly associated with aggressive tumor features, including AJCC tumor stage, vascular invasion, unclear surgical margin, viral hepatitis B infection, high CA19-9, and poor pre-surgical performance status. Moreover, BTC patients with low serum albumin had a shorter disease-free and long-term survival duration. 

Radical tumor resection (pR0) is prognostic for disease-free and overall survival after resection of biliary tract cancer [51,52]. The definition of pR0 varies globally, and pR0 is proposed to mean a tumor-free margin of ≥1 mm, according to the International Collaboration on Cancer Reporting (ICCR) [53]. Some studies showed that a margin width of 1.0 cm was associated with significantly improved survival in ICC [54,55,56]. Interestingly, our research found that only surgical margin <1 mm (R1) was significantly associated with poor recurrence-free and overall survival. In our study, the surgical margin of 1 mm had no survival difference compared with a surgical margin **≧** 1 mm. This result was similar to that of Shunsuke et al. 

The ANS can verify the groups with poor prognoses in BTC patients before surgery. In patients with ANS groups, 1 and 2 had more advanced tumor stages and shorter recurrence-free duration. 

### Strengths and Limitations of the Study

This study was strengthened by the inclusion of a large number of patients with long-term follow-ups in Taiwan—however, some limitations of our research merit further discussion. First, selection bias exists due to the retrospective study in a single medical center. Second, about 47% of BTC patients in this study did not receive lymph node dissection during the operation. Lymph node dissection was only recorded by the surgeon who operated, which may make it less objective. Patients spanned a wide range of years, during which surgical techniques and philosophies may change, and this change cannot be accurately assessed. In one real-world study in the United States, lymphadenectomy underperformed nationwide, and nodal clearance rarely met current guideline recommendations [57]. In our study, the disease-recurrence-free survival and overall survival in BTC patients without lymph node dissection did not differ from those who were lymph node-negative. Finally, there is no consensus on the optimal cut-off value in biliary tract cancer patients as a prognostic factor for albumin and NLR. Further studies are necessary to determine the optimal albumin and NLR cut-off values for prognoses in patients with BTC.

## 5. Conclusions

This study showed that the ANS with combined serum albumin and NLR could more accurately determine the prognosis of BTC patients receiving an operation. Patients with a higher ANS were found to have poorer survival outcomes and shorter tumor recurrence duration and were correlated with other poor prognostic factors. 

## Figures and Tables

**Figure 1 cancers-15-05474-f001:**
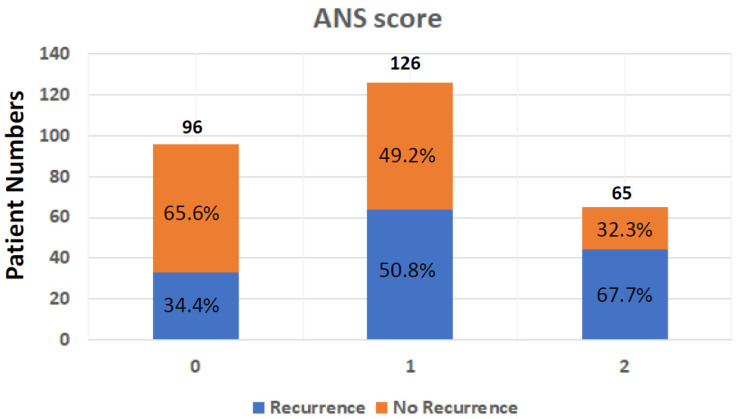
Tumor recurrence rate after operation in BTC patients from different ANS groups; ANS: albumin and neutrophil-to-lymphocyte ratio score; BTC: biliary tract cancer.

**Figure 2 cancers-15-05474-f002:**
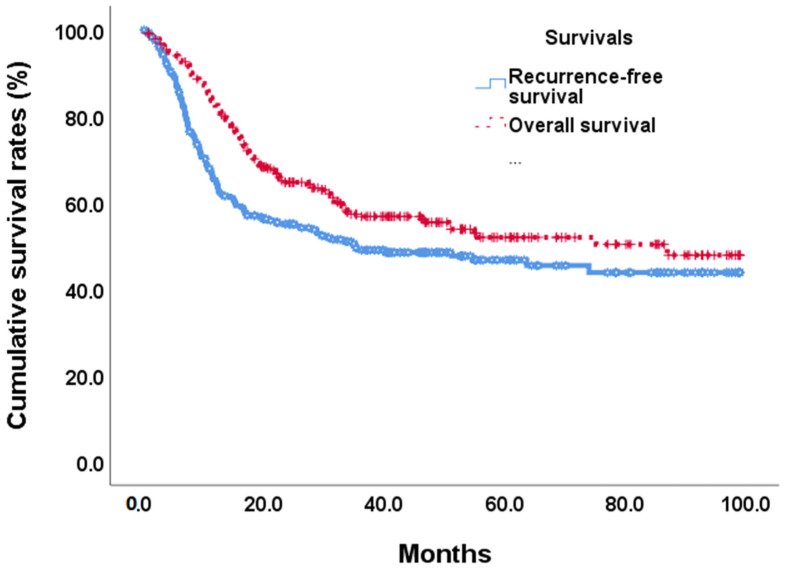
Overall survival and recurrence-free survival curves in BTC patients after tumor resection.

**Figure 3 cancers-15-05474-f003:**
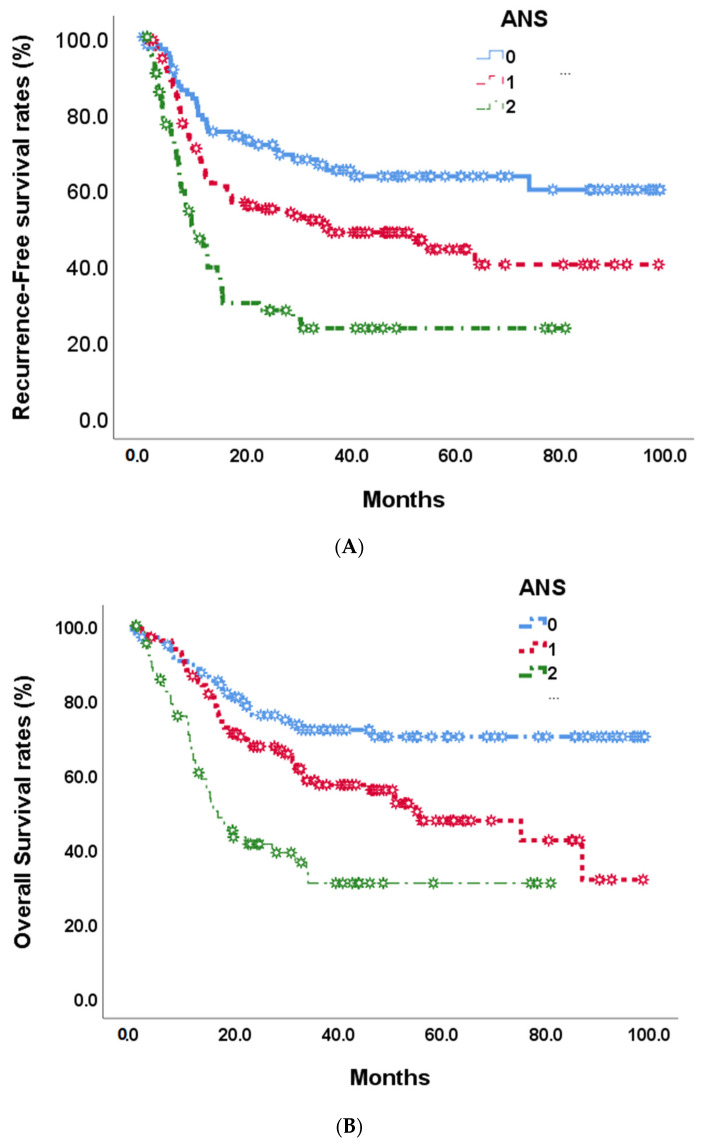
Recurrence-free survival (**A**) and overall survival (**B**) in 268 BTC patients by 3 preoperative ANS scores.

**Figure 4 cancers-15-05474-f004:**
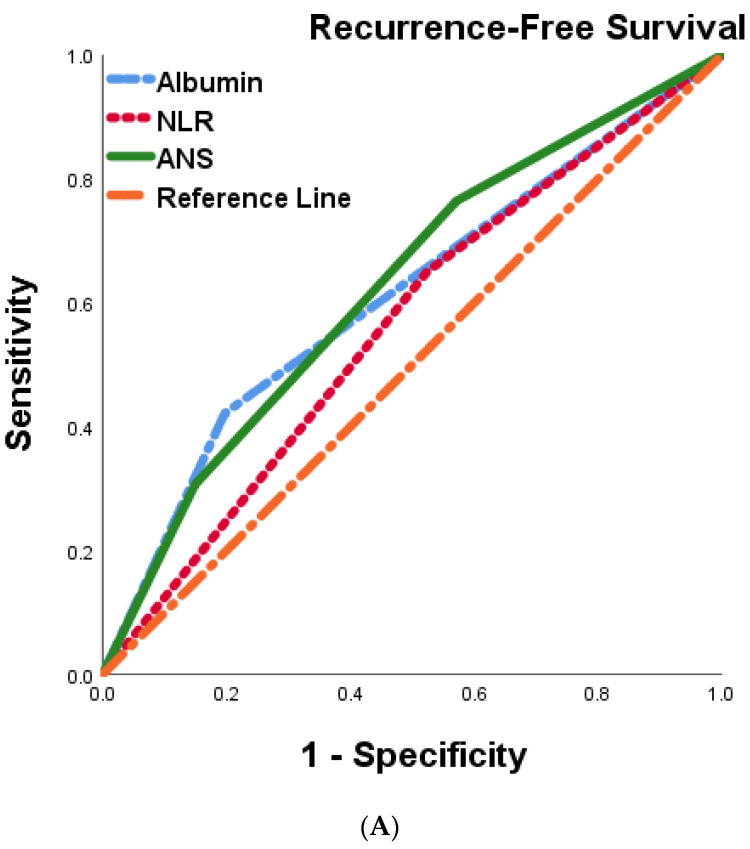
Comparison of the areas under the ROC curve for predicting recurrence-free survival (**A**) and overall survival (**B**) of 287 BTC patients undergoing curative resection.

**Table 1 cancers-15-05474-t001:** Clinical, pathological, and laboratory characteristics of the patients.

	Patient Numbers	Percentage (%)
**Gender**		
Male	167	58.2
Female	120	41.8
**Age (Median: 61.5** **± 10.93 years old)**		
<62	136	47.4
≧62	151	52.6
**Performance status (ECOG)**		
0	175	61.0
1	78	27.2
2	34	11.8
**Tumor site**		
Intrahepatic	177	61.7
Gall bladder	37	12.9
Common bile duct	51	17.8
Perihilar	22	7.7
**Tumor size**		
T1	81	28.2
T2	111	38.7
T3	51	17.8
T4	44	15.3
**Lymph nodes**		
N0	102	35.5
N1	47	16.4
N2	3	1
No dissection	135	47.0
**TMN stage**		
I	83	28.9
II	124	43.2
III	39	13.6
IV	41	14.3
**Hepatitis B**		
Yes	99	34.5
No	188	65.5
**Hepatitis C**		
Yes	48	16.7
No	239	83.3
**CA199 before surgery**		
<350	227	79.1
≧350	60	20.9
**Vascular invasion**		
No	133	46.3
Yes	134	46.7
Portal invasion	20	7.0
**Surgical margin**		
R0	217	75.6
R1	37	12.9
Close margin (1 mm)	33	11.5
**Adjuvant treatment**		
No	214	74.6
Chemotherapy	53	18.5
CCRT	13	4.5
Radiotherapy	7	2.4
**ERCC1**		
Negative	247	86.1
Positive	40	13.9
**XRCC1**		
Negative	251	87.5
Positive	36	12.5
**Albumin**		
<3.5	88	30.7
≧3.5	199	69.3
**NLR**		
<2.28	119	41.5
≧2.28	166	58.5

**Table 2 cancers-15-05474-t002:** Associations of albumin, NLR, and ANS with clinicopathological characteristics.

Clinical and Pathological Characteristics	Albumin			NLR			ANS			
	<3.5	≧3.5	*p*-Value	<2.28	≧2.28	*p*-Value	0	1	2	*p*-Value
	N = 88	N = 199		N = 119	N = 168		N = 96	N = 126	N = 65	
**Age**										
**<62**	**39**	97	0.489	63	73	0.113	52	56	28	0.260
≧62	49	102		56	95		44	70	37	
**Sex**										
Male	52	115	0.837	69	98	0.953	56	72	39	0.930
Female	36	84		50	70		40	54	26	
**Tumor location**										
Intrahepatic	46	131	0.132	85	92	0.012	70	76	31	0.037
Gall bladder	12	25		15	22		12	16	9	
Common bile duct	21	30		14	37		10	24	17	
Hilar	9	13		5	17		4	10	8	
**p-T Status**										
1–2	46	146	0.001	99	93	0.001	83	79	30	0.001
3–4	42	53		20	75		13	47	35	
**p-N status**										
Negative	29	73	0.001	37	65	0.001	32	46	24	0.001
Positive	26	24		12	38		7	22	21	
No dissection	33	102		70	65		57	58	20	
**AJCC pathological stage**										
I-II	53	154	0.003	98	109	0.001	82	88	37	0.001
III-IV	35	45		21	59		14	38	28	
**Vascular invasion**										
No	29	104	0.002	62	71	0.100	55	56	22	0.012
Yes	59	95		57	97		41	70	43	
**Surgical margin**										
R0	55	162	0.003	99	118	0.033	81	99	37	0.002
Close margin (1 mm)	16	17		8	25		7	11	15	
R1	17	20		12	25		8	16	13	
**DM.**										
No	64	143	0.88	94	113	0.029	76	85	46	0.150
Yes	24	56		25	55		20	41	19	
**HBV**										
No	68	120	0.005	63	125	0.001	50	83	55	0.001
Yes	20	79		56	43		46	43	10	
**HCV**										
No	76	163	0.351	92	147	0.023	75	105	59	0.108
Yes	12	36		27	21		21	21	6	
**CA19-9**										
<350	58	169	0.001	104	123	0.004	87	99	41	0.001
≧350	30	30		15	45		9	27	24	
**Performance status (ECOG)**										
0	40	135	0.002	83	92	0.026	73	72	30	0.001
1	33	45		23	55		13	42	23	
2	15	19		13	21		10	12	12	
**Recurrence**										
Yes	60	81	0.001	49	92	0.023	33	64	44	0.001
No	28	118		70	76		63	62	21	
**ERCC1**										
Negative	81	166	0.064	98	149	0.166	77	110	60	0.078
Positive	7	33		21	19		19	16	5	
**XRCC1**										
Negative	78	173	0.835	100	151	0.196	80	133	58	0.339
Positive	10	26		19	17		16	13	7	

**Table 3 cancers-15-05474-t003:** Univariate and multivariate analysis of RFS in patients with biliary tract cancer after treatment.

Variables	Category	Univariate			Multivariate		
		HR.	95% CI	*p*-Value	HR.	95% CI	*p*-Value
**Age**	<62 vs. **≧**62	0.937	0.67–1.31	0.699	0.762	0.53–1.09	0.141
Gender	Female vs. male	0.772	0.55–1.09	0.139	0.762	0.86–1.81	0.245
ECOG performance	0	Reference					
	1	1.56	1.08–2.28	**0.020**	1.159	0.78–1.73	0.473
	2	2.198	1.37–3.54	**0.001**	1.409	0.85–2.34	0.183
Tumor location	Intrahepatic	Reference					
	Gall bladder	0.728	0.41–1.31	0.288	0.541	0.29–1.02	0.057
	Common bile duct	1.40	0.92–2.13	0.114	1.285	0.75–2.20	0.359
	Hilar	1.52	0.86–2.68	0.148	0.476	0.25–0.92	**0.026**
p-T status	3–4 vs. 1–2	2.42	1.73–3.38	<0.001			
p-N status	Negative	Reference					
	Positive	2.35	1.54–3.61	**<0.001**			
	No lymph node dissection	1.089	0.74–1.61	0.667			
**AJCC pathological stage**	III–IV vs. I–II	2.525	1.80–3.55	**<0.001**	2.285	1.47–3.55	**<0.001**
**Vascular invasion**	Yes vs. No	2.048	1.45–2.90	**<0.001**	1.998	1.37–2.92	**<0.001**
**Surgical margin**	R0	Reference					
	Margin 1 mm	1.424	0.86–2.36	0.169	1.104	0.65–1.87	0.715
	R1	2.443	1.57–3.81	**<0.001**	1.510	0.92–2.47	0.101
**Adjuvant Treatment**	Yes vs. No	1.46	1.03–2.08	**0.035**	0.630	0.41–0.96	**0.032**
DM	Yes vs. No	0.902	0.62–1.30	0.580	0.853	0.58–1.26	0.426
HBV	Yes vs. No	0.620	0.43–0.90	**0.011**	0.669	0.44–1.03	0.067
HCV	Yes vs. No	0.819	0.52–1.30	0.399	1.121	0.68–1.85	0.654
CA19-9	**≧**350 vs. <350	3.373	2.36–4.82	**<0.001**	3.271	2.17–4.93	**<0.001**
ERCC1	Positive vs. negative	1.244	0.74–2.09	0.412	0.837	0.51–1.38	0.484
XRCC1	Positive vs. negative	1.16	0.71–1.90	0.566	1.27	0.75–2.15	0.372
Albumin	**≧**3.5 vs. <3.5	0.386	0.28–0.54	**<0.001**			
NLR	**<2.28 vs.** **≧2.28**	1.615	1.14–2.29	**0.007**			
ANS	2	Reference					
	0	0.290	0.18–0.46	**<0.001**	0.497	0.30–0.82	**0.007**
	1	0.494	0.34–0.73	**<0.001**	0.638	0.42–0.96	**0.031**

**Table 4 cancers-15-05474-t004:** Univariate and multivariate analysis of OS in patients with biliary tract cancer after treatment.

Variates	Category	Univariate			Multivariate		
		HR	95% CI	*p*-Value	HR	95% CI	*p*-Value
**Age**	<62 vs. **≧**62	0.761	0.53–1.09	0.134	1.005	0.69–1.47	0.980
Gender	Female vs. male	0.904	0.63–1.30	0.587	1.071	0.72–1.59	0.732
ECOG performance	0	Reference					
	1	2.151	1.45–3.19	**<0.001**	1.502	0.98–2.30	0.062
	2	2.714	1.64–4.49	**<0.001**	1.623	0.94–2.79	0.080
Tumor location	Intrahepatic	Reference					
	Gall bladder	0.968	0.52–1.79	0.866	0.541	0.45–1.66	0.665
	Common bile duct	1.640	1.06–2.53	**0.026**	1.466	0.84–2.57	0.183
	Hilar	1.928	1.06–3.50	**0.031**	0.819	0.42–1.61	0.561
p-T status	3–4 vs. 1–2	2.551	1.79–3.65	**<0.001**			
p-N status	Negative	Reference					
	Positive	2.686	1.63–4.29	**<0.001**			
	No lymph node dissection	1.126	0.74–1.73	0.586			
**AJCC pathological stage**	III–IV vs. I–II	2.441	1.70–3.51	**<0.001**	2.140	1.36–3.38	**0.001**
**Vascular invasion**	Yes vs. no	2.303	1.57–3.37	**<0.001**	1.998	1.33–3.01	**0.001**
**Surgical margin**	R0	Reference					
	Margin 1 mm	1.591	0.93–2.73	0.092	1.148	0.63–2.03	0.635
	R1	3.623	2.37–5.55	**<0.001**	2.240	1.40–3.57	**0.001**
**Adjuvant Treatments**	Yes vs. No	1.419	0.97–2.07	0.070	0.622	0.40–0.97	**0.034**
DM	Yes vs. No	1.436	0.99–2.09	0.060	1.106	0.74–1.65	0.624
HBV	Yes vs. No	0.516	0.34–0.78	**0.002**	0.675	0.42–1.08	0.103
HCV	Yes vs. No	0.760	0.46–1.27	0.294	1.017	0.59–1.76	0.951
CA19-9	**≧**350 vs. <350	2.875	1.98–4.19	**<0.001**	2.333	1.54–3.54	**<0.001**
ERCC1	Positive vs. negative	0.972	0.58–1.62	0.913	0.861	0.51–1.45	0.573
XRCC1	Positive vs. negative	0.74	0.45–1.22	0.24	0.652	0.39–1.08	0.097
Albumin	<3.5 vs. **≧**3.5	0.385	0.27–0.55	**<0.001**			
NLR	**≧2.28 vs. <2.28**	1.751	1.20–2.56	**0.004**			
**ANS**	2	Reference					
	0	0.261	0.16–0.43	<0.001	0.532	0.31–0.93	**0.027**
	1	0.482	0.32–0.73	<0.001	0.626	0.41–0.97	**0.034**

**Table 5 cancers-15-05474-t005:** Areas under the receiver operating characteristic curves for prognostic indexes of albumin, NRL, and ANC for predicting postoperative survival in 287 BTC patients undergoing curative resection.

	Recurrent Free Survival		Overall Survival	
Variates	Area under the ROC Curve (95% CI)	*p*-Value	Area under the ROC Curve (95% CI)	*p*-Value
**ANC**	0.627 (0.563–0.691)	<0.001	0.636 (0.571–0.0700)	<0.001
**Albumin**	0.612 (0.547–0.677)	0.001	0.611 (0.544–0.678)	0.001
**NLR**	0.536 (0.497–0.629)	0.065	0.575 (0.509–0.642)	0.029

## Data Availability

Data are contained within the article.

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
