# Peer review of "Combined Serum ALBUMIN with Neutrophil-to-Lymphocyte Ratio Predicts the Prognosis of Biliary Tract Cancer after Curative Resection"

_cancers, 2023, doi:10.3390/cancers15225474_

Round 1

Reviewer 1 Report

Comments and Suggestions for Authors

This is an interesting analysis of patients with biliary tract cancer who underwent resection. Retrospectively, the easily available laboratory information of serum albumin and NLR were assessed and appear to risk stratify patients for long term outcomes. This is important in that many patients with biliary cancers are diagnosed in low medical resource areas of the world and this stratification could be important in surgical planning.  The primary analyses appear to be methodologically sound.  However, I would recommend some adjustment to conclusions. Lines 313-318 overstate the conclusions that can be drawn on the basis of this analysis.  While these prognostic groups are appropriate to study for the role of nutrition supplementation or preoperative chemotherapy, it is not yet clear that these should be treatment recommendations on the basis of the data.  Moreover, surgical margin recommendations are not germane to this report.  Finally, the paragraph in lines 320-325 is internally inconsistent (same results or did not predict survival?) and should be reworked for clarity or deleted.

Comments on the Quality of English Language

In general, the English is fine. I am curious if paragraph 320-325 doesn't make sense due to translation issues.

Author Response

Nov. 04, 2023

Dear Editors:

We appreciate very much your kind consideration as regards revising the manuscript entitled "Combined serum albumin with neutrophil-to-lymphocyte ratio predicts the prognosis of biliary tract cancer after curative resection” (cancers-2718707).

This manuscript has been re-evaluated according to the comments given by your panel of reviewers. In order to reply to the specific comments of the reviewers, we have performed all of the suggested experiments and revised the manuscript. We also made a point-by-point response to each comment made by the reviewers, which are all contained in the attached "Response to the Reviewers".

Lastly, we would like to thank you once again for providing us the opportunity to improve our manuscript. We hope that these revisions are adequate and that the manuscript is now acceptable for publication in Cancer Cell International.

Sincerely yours,

Yen-Yang Chen

Point-by-point responses to reviewer's comments.

RE: " Combined serum albumin with neutrophil-to-lymphocyte ratio predicts the prognosis of biliary tract cancer after curative resection ".

We would like to thank the reviewers for the thorough reading of our manuscript as well as their valuable comments. We have followed their comments closely and feel that their suggestions have further strengthened the manuscript. Below are our point-by-point responses.

Response to Reviewer 1's Comments:

  1. Lines 313-318 overstate the conclusions that can be drawn on the basis of this analysis. While these prognostic groups are appropriate to study for the role of nutrition supplementation or preoperative chemotherapy, it is not yet clear that these should be treatment recommendations on the basis of the data. Moreover, surgical margin recommendations are not germane to this report. Finally, the paragraph in lines 320-325 is internally inconsistent (same results or did not predict survival?) and should be reworked for clarity or deleted.

Response: Thanks for your very kind comments.

  1. We rewrite Lines 313-318 in the manuscript to delete the overstated part. (Before tumor resection, we should improve their nutritional status and consider more effective treatments, such as neoadjuvant chemotherapy or concurrent chemoradiotherapy, to ameliorate cancer-induced inflammation.)
  2. Moreover, surgical margin recommendations are not germane to this report: we agree with your comment and delete this sentence. (Besides, the surgeons should take care of surgical tumor margins of at least more than 1mm. After the operation, those patients need closer follow-up and aggressive adjuvant treatments).
  3. Finally, the paragraph in lines 320-325 is internally inconsistent (same results or did not predict survival?) and should be reworked for clarity or deleted. We agree with your comment and delete these parts.

Reviewer 2 Report

Comments and Suggestions for Authors

1. line 21-22 in the annotation - text repetition

2. Figure 1 repeats the data given in Table 2.

3. I wonder if we divide the entire study group into those who had a relapse and those who did not, and compare the main characteristics of the groups with each other, how will albumin and NLR levels change in these groups? Maybe in these groups it would be possible to determine the cut-off points differently and get an even more accurate forecast, in your opinion?

Author Response

Nov. 04, 2023

Dear Editors:

We appreciate very much your kind consideration as regards revising the manuscript entitled "Combined serum albumin with neutrophil-to-lymphocyte ratio predicts the prognosis of biliary tract cancer after curative resection” (cancers-2718707).

This manuscript has been re-evaluated according to the comments given by your panel of reviewers. In order to reply to the specific comments of the reviewers, we have performed all of the suggested experiments and revised the manuscript. We also made a point-by-point response to each comment made by the reviewers, which are all contained in the attached "Response to the Reviewers".

Lastly, we would like to thank you once again for providing us the opportunity to improve our manuscript. We hope that these revisions are adequate and that the manuscript is now acceptable for publication in Cancer Cell International.

Sincerely yours,

Yen-Yang Chen

Point-by-point responses to reviewer's comments.

RE: " Combined serum albumin with neutrophil-to-lymphocyte ratio predicts the prognosis of biliary tract cancer after curative resection ".

We would like to thank the reviewers for the thorough reading of our manuscript as well as their valuable comments. We have followed their comments closely and feel that their suggestions have further strengthened the manuscript. Below are our point-by-point responses.

Response to Reviewer 2's Comments:

  1. Line 21-22 in the annotation - text repetition

Response: We agree with this valuable comment. We rewrite, “In this study, we investigated the possibility of combining serum albumin concentrations and neutrophil-to-lymphocyte ratios (NLR) to create a score as ANS to predict the prognoses of biliary tract cancer before surgery”.

  1. Figure 1 repeats the data given in Table 2.

Response: We agree with this comment. In Figure 1, we tried to make a more simple picture to let the audience easily figure out the complicated table 2. In Figure 1, the difference in the recurrence rate is clearly shown in ANS 0, 1, and 2.

3. I wonder if we divide the entire study group into those who had a relapse and those who did not, and compare the main characteristics of the groups with each other, how will albumin and NLR levels change in these groups? Maybe in these groups it would be possible to determine the cut-off points differently and get an even more accurate forecast, in your opinion?.

Response: We think this is a very important opinion.

The cut-off value of albumin 3.5 is used daily to determine the nutritional status of cancer patients.

The cut-off value of NLR was according to characteristic (ROC) curve analyses.

The albumin value and NLR also showed significant differences in recurrence-free survival rates in the Kaplan-Meier survival analyses.
